# Neural Sensitivity to Mutual Information in Intermediate-Complexity Face Features Changes during Childhood

**DOI:** 10.3390/brainsci9070154

**Published:** 2019-06-28

**Authors:** Benjamin Balas, Assaf Harel, Amanda Auen, Alyson Saville

**Affiliations:** Department of Psychology, Center for Visual and Cognitive Neuroscience, North Dakota State University, Fargo, ND 58102, USA

**Keywords:** face detection, visual development, electrophysiology

## Abstract

One way in which face recognition develops during infancy and childhood is with regard to the visual information that contributes most to recognition judgments. Adult face recognition depends on critical features spanning a hierarchy of complexity, including low-level, intermediate, and high-level visual information. To date, the development of adult-like information biases for face recognition has focused on low-level features, which are computationally well-defined but low in complexity, and high-level features, which are high in complexity, but not defined precisely. To complement this existing literature, we examined the development of children’s neural responses to intermediate-level face features characterized using mutual information. Specifically, we examined children’s and adults’ sensitivity to varying levels of category diagnosticity at the P100 and N170 components. We found that during middle childhood, sensitivity to mutual information shifts from early components to later ones, which may indicate a critical restructuring of face recognition mechanisms that takes place over several years. This approach provides a useful bridge between the study of low- and high-level visual features for face recognition and suggests many intriguing questions for further investigation.

## 1. Introduction

An important way to understand how face recognition develops is to characterize the information used by the developing visual system to categorize faces or non-faces. Faces are complex visual stimuli, which means that there are a lot of visual features that one could use to detect or recognize a face, but some of these features are more useful than others. Limiting observers’ access to specific information in face images can have substantial effects on face recognition in many tasks, revealing what information is critical to face detection or identification at different points in development. To date, there have been many developmental studies (including both behavioral and neural responses) examining how adults’ and children’s face recognition is affected by the presence of specific low-level features (spatial frequency or orientation sub-bands), and the availability of higher-level visual information (configural or holistic face appearance). Each approach has yielded useful insights regarding how mechanisms for face recognition are tuned to diagnostic information at different stages of development. In the current study, we argue that an important complement to these two avenues of research is to examine how intermediate representations of facial appearance also change developmentally. Mid-level face features occupy a useful middle ground between low- and high-level descriptors of face structure and allow for a balance between feature complexity and our ability to objectively quantify the information carried by features under consideration. We elaborate on these points below by briefly describing several main results concerning how sensitivity to low-level and high-level features for face recognition develops during childhood, then describing our approach to studying mid-level representations of facial structure using event-related potentials (ERPs).

With regard to low-level visual information, adults [1,2,3,4] and children [5,6] tend to perform better across a range of face recognition tasks when intermediate spatial frequency information is available. Depending on the task, children either exhibit adult-like sensitivity to spatial frequency fairly early in childhood [7] or gradually develop the same profile during middle childhood [6]. The orientation content of face images also matters a great deal: In many tasks, horizontally-oriented features support better performance in adults than vertically-oriented features [8,9], though there are cases in which vertical orientation energy is more useful [10]. Children also exhibit a similar reliance on horizontally-oriented features for various face recognition tasks [11], and this bias also appears to emerge gradually over development [12,13]; see de Heering et al. [14] for results demonstrating that infants exhibit sensitivity to horizontal orientation information]. In both cases, we see clear evidence that specific visual features known to be computed at relatively early stages of the visual processing vary in their utility for face recognition tasks, and adult-like sensitivity to this variation develops during childhood.

Besides these low-level features, there is also substantial evidence that higher-level visual features vary in their efficacy for face recognition tasks and that children develop sensitivity to this variation, too. For example, converging evidence from multiple tasks suggests that adult and child observers tend to use the eye region for face recognition tasks more than the mouth or nose [15,16,17]. This also varies somewhat across tasks, but the larger point is that this is another example of the visual system relying more heavily on one class of visual features than another. In this case, the specific visual information that is being used is harder to describe precisely, but nonetheless, these studies demonstrate that among more complex visual features, there are still some features that are better than others for face processing. At perhaps the highest level of complexity, it is well-established that face recognition is to some ‘holistic,’ in that it relies in some crucial way on the entire face pattern rather than parts of the face [18]. This is frequently demonstrated using testing paradigms including the composite face effect (or CFE, [19]), face inversion [20,21], or the part-whole effect [22,23], all of which have been shown to have a developmental trajectory that continues to unfold during childhood to varying degrees [24,25,26,27,28,29]. Again, the developing visual system acquires adult-like sensitivity to diagnostic face information during childhood.

Limiting ourselves to either a low-level vocabulary of visual features (spatial frequencies, orientation passbands) or a high-level vocabulary (holistic face appearance) comes with an inherent trade-off, however: As the complexity of our feature vocabulary increases, the precision with which we can speak about the visual information being used decreases. The nature of this trade-off limits our ability to develop a comprehensive picture of how face recognition develops in terms of information use during childhood. Low-level features are easy to specify computationally, but we are confident the visual system ultimately does more to process faces than just measure these basic features of appearance. We still do not really know how low-level features are combined into something more complex in service of recognition tasks. Without that, we are missing a crucial aspect of what is changing as face recognition develops. Conversely, high-level features may be more relevant to these later stages of face processing, but there remains no widely accepted model of what holistic or configural face processing is in terms of specific computations. Without that, we are missing a different, crucial aspect of what is changing as face recognition develops. We suggest that an important complement to both of these lines of research is to examine intermediate-complexity features for face categorization [30]. Ideally, we would like to identify an approach that allows us to increase the complexity of the feature vocabulary under consideration while retaining a computational description of how these features differ from one another. If possible, examining such mid-level representations for face and non-face categorization developmentally would be an important way to bridge the gap between low-level and high-level characterizations of the development of face recognition.

In the current study, we examined children’s developing sensitivity to mid-level features for face recognition by using mutual information (or MI) to quantify the utility of fragments of face and non-face images for categorization. This approach was first introduced by Ullman, Vidal-Naquet, & Sali [31] to identify visual features of intermediate complexity that could be used for basic-level categorization. Briefly, mutual information is a quantity that reflects the diagnosticity of a particular image fragment for an object category: How likely are we to find this fragment in images that depict the target category, and how likely are we to find the same fragment in images that do not? For a face fragment to have high mutual information, it must be the case that we are quite likely to find such fragments in images of faces and unlikely to find it in non-face images. Face fragments with high MI tend to be easier to categorize as faces for both adult [32,33] and child observers [34], which suggests that mutual information is a perceptually meaningful quantity. Further, face fragments with high MI also tend to be neither very large, nor very small, nor do they tend to capture easily labeled portions of a face image. This suggests that MI may be a good tool for examining the nature of intermediate-level representations of face appearance; Fragments with high MI are more complex than low-level features, but not as complex as ‘holistic’ or ‘configural’ representations of the face.

Here, we used event-related potentials (ERPs) to examine children’s neural sensitivity to image fragments of varying MI drawn from a face and a non-face category (cars). Specifically, we examined the response properties of the P100 and the N170, both of which are well-established face-sensitive ERP components [35,36] that are measurable in adult [37,38] and child populations [39]. By “face-sensitive,” we mean that both of these components have response properties that suggest that they reflect some aspect of face processing in particular rather than more general object recognition mechanisms. For our purposes, it is particularly important that both of these components have been found to reflect sensitivity to low-level and high-level aspects of facial appearance in a manner that is consistent with the behavioral studies described above. Both children’s and adult’s N170 components are sensitive to the orientation energy [40,41] and spatial frequency content [42] of face images, for example, though children’s sensitivity is not adult-like until late in middle childhood or adolescence. In terms of high-level visual information, adults’ N170 response to schematic “smiley” faces [43], the composite face illusion [44], and other manipulations of holistic face appearance [45,46] suggests that these components also reflect tuning to face features with higher complexity (for a review and discussion, see Reference [47]). The N170, which is known to have a larger response to faces than other object categories, also varies with stimulus properties like orientation, spatial frequency content, and holistic structure, in a manner that appears to reflect the behavioral signatures of face-specific processing. Likewise, though the P100 is known to be sensitive to low-level stimulus properties including contrast and luminance, the P100 also exhibits an inversion effect for face stimuli, suggesting that it too reflects the activity of mechanisms for face recognition. In both cases, because these components tend to vary their response properties as a function of how face-like a stimulus is, we anticipate that varying the informativeness of a face fragment will also vary the “face-ness” of an image and therefore the responses of the P100 and N170. Further, with regard to development, children’s neural responses to variations of holistic face appearance are slow to reach adult-like levels [48], suggesting that during childhood, these mechanisms are perhaps not as selective for faces or as sensitive to specific visual features that define face images as a class. Because children’s neural sensitivity to both low-level and high-level information for face recognition develops in a manner that is measurable with ERP, we may also ask about the neural response to intermediate-level features. Adults’ ERP responses are sensitive to the informativeness of face fragments as operationalized via MI: Parametrically varying fragment informativeness results in a gradient-like amplitude change in a negative-going component peaking around 270 ms [32]. That is, mutual information appears to modulate an N270 component in adult participants, as observed by Harel et al. in previous work [32]. More generally, mid-level features are sufficient to drive responses in object-selective parts of the ventral visual stream [49,50,51]. To our knowledge, however, little work has been done to measure how children’s neural responses may develop to reflect sensitivity to varying levels of diagnosticity in intermediate-level face or non-face object fragments. We ask, therefore, whether children’s P100 or N170 components reflect sensitivity to face MI during childhood, and also whether they also exhibit an N270 component in response to face fragments that similarly reflects sensitivity to varying levels of fragment informativeness.

Our primary hypotheses were that (1) Children may be less sensitive to varying levels of MI than adults, reflecting the ongoing refinement of feature vocabularies for categorization. (2) The onset of adult-like levels of MI sensitivity for faces and objects may differ, reflecting different developmental time courses for the acquisition of robust mechanisms for face and non-face categorization. We investigated these hypotheses by measuring ERP responses in school-age children (6–10 years) and adults while they completed a 2AFC categorization task using face and car fragments of varying levels of MI. In particular, we examined how the amplitudes of the P100 and N170 were affected by MI as a function of category and participant age.

## 2. Methods

### 2.1. Participants

We recruited a total of 48 participants for this experiment. Specifically, we recruited 16 participants (12 female, 4 male) between the ages of 5–7 years old (M = 6;3, S.D. = 0.77 years), 16 participants (10 female, 6 male) between the ages of 8–10 years old (M = 8;7 years, S.D. = 0.80 years) and 16 adult participants (12 females, 4 males) between the ages of 18–23 years old (M = 19;3 years, S.D. = 1.3 years). All participants self-reported normal or corrected-to-normal vision and no history of visual or neurological disorders. Further, all participants were assessed to be right-handed according to the Edinburgh Handedness Inventory [52].

Adult participants were recruited from the NDSU Psychology Undergraduate Study Pool and received course credit for their participation. Child participants were recruited from the greater Fargo-Moorhead area and received a small monetary amount and a book of their choosing for their participation. We obtained written informed consent from all adult participants and from the parent/guardian of each child participant. Children over the age of 7 years old also provided independent written assent to participate. All procedures for recruitment and testing were approved by the NDSU IRB, in accordance with the principles outlined in the Declaration of Helsinki.

### 2.2. Stimuli

Our stimulus set was comprised of grayscale images depicting portions (“fragments”) of faces (96 images) or cars (96 images). Within each object category, these fragments varied in terms of the mutual information that they provided about object category based on their appearance. This term refers to a quantity that describes how diagnostic a particular fragment is of the target object category. For example, a fragment with high mutual information (or MI) would be very likely to appear in the target category (e.g., faces) but unlikely to appear in a non-target category (e.g., cars). Details regarding how MI is calculated for individual face fragments given a large set of images from multiple categories can be found in Ullman, Vidal-Naquet & Sali [31] and Harel et al. [32], but we provide a brief overview of how MI is defined for an individual fragment given a set of images in which the target category either does or does not appear in each image. In both References [31] and [32], the mutual information equation is applied to quantify the information that an individual fragment provides about a particular object class:(1)I(C,F)=H(C)−H(C|F)

Here, I(C,F) is our measure of mutual information, and H refers to entropy:(2)$$H\left(x\right)=-\sum_{i=1}^{x}p\left(x_{i}\right)\log\left(p\left(x_{i}\right)\right)$$

Entropy, as defined above, quantifies the extent to which class uncertainty is reduced by the observation of a particular fragment. In the case of a candidate grayscale and a set of images, we estimate the relevant quantities by attempting to detect the fragment in images that do or do not contain an instance of the object class. Specifically, we use cross-correlation to determine whether or not each image in a training set contains instances of the fragment: if the cross-correlation exceeds a threshold value in a particular image, then we consider that a successful detection. Applying this procedure across all of the images in the training set makes it possible for us to calculate the frequency with which a fragment occurred in images containing the target category and images that lack the target category. These frequencies are then used to compute mutual information as defined above.

For the purposes of the present study, the key feature of our stimulus set is that this property varies across images in a quantifiable way. Some of our fragments are highly diagnostic of their category (face or car), while others are less so. We divided face and car fragments into three non-overlapping categories based on MI values to obtain 32 “low,” medium,” and “high” images in each category (Figure 1). Image fragments within these categories varied in size and position with regard to the larger image from which they were taken. We note that this partitioning of the stimulus set represents a subset of the stimuli used in Reference [32], which were binned into five MI levels according to ranked MI values. We included image fragments from their lowest MI category, their highest MI category, and an intermediate category. As such, we attempted to present participants with fragments that varied substantially in terms of their informativeness.

### 2.3. Procedure

We asked participants to complete a 2AFC face/car categorization task using the image fragments described above while we recorded continuous EEG from scalp electrodes. We measured EEG from all participants using 64-channel Hydrocel Geodesic Sensor Nets by EGI, which were connected to an EGI 400 NetAmps amplifier. Prior to the testing session, we measured the head circumference of each participant to determine an appropriately sized sensor net and marked the target position of the vertex electrode on the scalp with a red grease pencil. The sensor net was then soaked in a solution of KCl for approximately 5 min, after which the net was applied to the participant’s scalp. Next, we re-seated and/or applied additional KCl solution to individual electrodes to establish stable impedances below 25 kΩ. During EEG recording, continuous EEG was referenced to the vertex electrode with a sampling rate of 250 Hz and filtered online with a 0.1 Hz high-pass filter.

Participants completed the categorization task seated in front of a 1024 × 768 LCD monitor that was installed in an electrically-shielded and sound-attenuated room. Participants were seated approximately 40 cm from the display, and at this distance individual stimuli subtended between 4–10 degrees of visual angle depending on the size of individual fragments. During the task, participants were presented with individual fragments from each target category in a pseudorandomized order. Each image was presented for 500 ms against a white background, after which participants had unlimited time to label the image as either a “face” or a “car” using their right or left thumbs to push the corresponding buttons on a custom button box. Half of our participants in each age group held the button box in a “flipped” orientation so that the assignment of category label to hand was balanced across participants. Following their response, we included an intertrial interval that varied between 500 ms–1000 ms from trial to trial subject to random draws from a uniform distribution bounded by these values. Each stimulus was presented one time during the testing session for a total of 192 trials in the entire task. Participants tended to complete the testing session in about 15 min. All stimulus presentation and response collection routines were carried out using custom routines written using EPrime v. 2.0 (PST, Pittsburgh, PA, USA). All EEG recording and event marking routines were carried out using NetStation v. 5.0 (EGI, Eugene, OR, USA).

## 3. Results

### 3.1. Behavioral Results

For each participant, we calculated the proportion of correct responses to face and car fragments at each MI level (see Figure 2 below). The behavioral data files from two participants (one 5–7 year old and one 8–10 year old) were corrupted and were thus excluded from this analysis. We analyzed these data using a 2 × 3 × 3 mixed-design ANOVA with category (face vs. car) and MI level (low, medium, and high) as within-subject factors and age group (5–7 year-olds, 8–10 year-olds, and adults) as a between-subjects factor. This analysis revealed a main effect of MI level (*F*(2,88) = 42.08, *p* < 0.001, partial η^2^ = 0.49) and a main effect of age group (*F*(2,44) = 6.53, *p* = 0.003, partial η^2^ = 0.23). The main effect of MI level was the result of significant differences between performance with high MI fragments relative to both low (*t* = 8.74, *p* < 0.001) and medium MI fragments (*t* = 10.52, *p* < 0.001). The main effect of group was the result of significant differences between adults’ performance and both 5–7 year-olds (*t* = 3.2, *p* = 0.008) and 8–10 year-olds (*t* = 2.99, *p* = 0.014). No other main effects or interactions reached significance.

### 3.2. EEG Results

We examined both the P100 ERP component and the N170 component subject to variation in the image category (face vs. car), MI level (low, medium, or high) and age group (5–7 year-olds, 8–10 year-olds, adults). In each case, we characterized components in terms of their mean amplitude during time windows determined independently for each participant group. We chose not to pursue an analysis of peak latency for either component for two main reasons: (1) Due to the difficulty in defining meaningful peak latency values for our child participants. While adult participants tended to have well-defined unimodal P100 and N170 components, several participants in both child age groups exhibited multimodal or broad P100 and N170 morphologies. In these cases, peak latency is extremely difficult to meaningfully define, because the data contain multiple maxima/minima within the time window chosen for component analysis. (2) Prior work by Harel et al. [32] using intermediate-level fragments to examine ERPs also only reported results regarding component amplitude. As such, we chose to focus on the mean amplitude because it is an easy-to-measure and robust means of characterizing ERP components [53]. While this limits the scope of our analysis somewhat, we think it also allows us to focus our discussion on a key set of results. Because several of the child participants exhibited multimodal responses, we focus our analysis on mean amplitude measures that do not rely on identifying a single peak. We continue by describing our processing pipeline, followed by a description of our results for each component.

### 3.3. EEG Pre-Processing

All EEG pre-processing steps were carried out using the NetStation v5.0’s NetStation Tools interface. For each participant, we began by applying a 30 Hz low-pass filter to the continuous EEG data recorded at each electrode. We continued by segmenting each participant’s data by condition using stimulus onset triggers inserted into the continuous EEG data by EPrime. We bounded these segments with a pre-stimulus onset period of 100 ms and a post-stimulus period of 1000 ms, yielding segments that were 1100 ms in total length. Following segmentation, we applied a baseline correction to each segment by subtracting the average value measured during the 100 ms pre-stimulus period at each sensor from the remaining 1000 ms of the segmented trial. Following this baseline correction routine, we applied NetStation’s artifact detection algorithms to identify and remove trials contaminated by eye blinks and saccades. Automatic artifact detection was also confirmed by independent examination of each subject’s data by two members of the research team. Finally, we applied NetStation’s bad channel replacement algorithm, which replaces channels that exceed a fixed proportion of bad trials with data interpolated from nearby sensors using spherical interpolation. We subsequently calculated an average ERP for each participant by averaging segments together within the subject category at each sensor.

### 3.4. ERP Component Analysis

We identified our target ERP components by examining the grand average ERP calculated across all participants in each age group. Within each group, we calculated the average ERP across all conditions and examined both the topography and timecourse of deflections corresponding to the P100 and N170 to determine sensors of interest and time windows for each component. In all groups, we chose to characterize both the P100 and N170 using sensors 29, 30, and 32 in the left hemisphere and sensors 43, 44, and 47 in the right hemisphere (these correspond to electrodes T5 and T6 in the standard 10–20 system, along with two adjacent electrodes in each hemisphere). With regard to time windows for analyzing our target components, we chose to examine the P100 using the following time windows for each age group: 5–7 year-olds: 96–272 ms; 8–10 year-olds: 96–272 ms; Adults: 96–156 ms. We chose to examine the N170 using the following time windows: 5–7 year-olds: 172–256 ms; 8–10 year-olds: 168–244 ms; Adults: 156–204 ms. We note that the choice of defining different component time windows for each age group was motivated by the observation that children’s ERP components are frequently slower in latency than adults and also often broader [39]. Faced with this variability as a function of age, defining a standard time interval for each component is unlikely to capture the full morphology of a given component across different participants. Thus, we have opted to use age-group specific time intervals to ensure that we adequately describe each component across conditions. In Figure 3a,b, we display grand average ERPs for each condition measured over left and right hemisphere electrodes.

Within each of these time windows, we characterized each component using the mean amplitude measured within that interval, averaged across the three sensor groupings in the left and right hemisphere independently. We examined the effects of image category, mutual information, and age group by analyzing the values we obtained for each descriptor via a 2 × 2 × 3 × 3 mixed-design ANOVA with hemisphere, target category and MI level as within-subject factors and age group as a between-subjects variable. Our key prediction was that sensitivity to face MI level (but not car MI level) would increase with age, leading to a 3-way interaction between target category, MI level, and age group. In Figure 4, we display the average amplitudes measured for both components as a function of these factors.

### 3.5. P100 Amplitude

Our analysis of the P100 mean amplitude (see Table 1 for the full ANOVA table) revealed a significant main effect of MI level (*p* = 0.017) and a marginal effect of age group (*p* = 0.079). The former was the result of larger P100 amplitudes to high MI fragments compared to low MI fragments (*t* = 2.89, Cohen’s d = 0.41, *p* = 0.013), while the latter was the result of a trend for larger amplitudes for younger participants, which is consistent with known differences between child and adult EEG signal-to-noise.

These main effects were qualified by a number of significant interactions, including two-way interactions between MI level and category (*p* = 0.030) and MI level and hemisphere (*p* = 0.005). The former interaction was qualified by a three-way interaction between object category, MI level, and age group (*p* = 0.027). In the interests of streamlining our discussion with regard to the outcomes that are critically important to our developmental hypotheses, we will focus here on further understanding this three-way interaction.

To examine how varying MI levels differentially impacted P100 amplitudes as a function of age group and target category, we carried out tests of simple main effects for MI within each combination of age group and category, collapsing across the hemisphere. These tests revealed that while face MI level led to significantly different mean amplitudes for the P100 in our youngest age group, (*F*(2,30) = 5.87, *p* = 0.007), neither older children (*F*(2,30) = 1.89, *p* = 0.17) nor adults (*F*(2,32) = 0.69, *p* = 0.57) exhibited such an effect. Furthermore, the MI level of car fragments did not lead to significantly different amplitudes in any of the three age groups. Thus, we suggest that the interaction we observed between category, MI level, and age category was largely driven by the sensitivity of the P100 to face MI level in young participants.

### 3.6. N170 Amplitude

Our analysis of the N170 amplitude (see Table 2 for full ANOVA results) revealed a significant main effect of the target category (*F*(1,46) = 13.46, *p* < 0.001), but no other significant main effects. We also observed a significant 3-way interaction between target category, hemisphere, and age group (*F*(2,46) = 3.54, *p* = 0.037) and a marginally significant 3-way interaction between category, MI level, and age group (*F*(4,92) = 2.36, *p* = 0.059). Though this latter effect did not reach the standard threshold for significance in an NHST framework, but because it is directly relevant to our predictions we continue by further exploring this interaction using simple main effects analysis.

As we did with the P100 data, we further examined how varying MI levels differentially impacted N170 amplitudes as a function of age group and target category via tests of simple main effects for MI within each combination of age group and category, collapsing across the hemisphere. These tests revealed that face MI level led to significantly different mean amplitudes for the N170 in adult participants, (*F*(2,32) = 4.2, *p* = 0.020), but neither younger children (*F*(2,30) = 1.86, *p* = 0.17) nor older children (*F*(2,30) = 0.72, *p* = 0.50) exhibited such an effect. Additionally, as before, the MI level of car fragments did not lead to significantly different amplitudes in any of the three age groups. In this case, we, therefore, suggest that the marginally significant interaction we observed between category, MI level, and age category may reflect face-specific sensitivity to MI at the adult N170 that is not observable at younger ages.

## 4. Discussion

There are several interesting features of our results that support our initial hypothesis that there are ongoing changes in how mid-level visual features are used for face recognition during middle childhood. Our key hypothesis regarding children’s sensitivity to MI relative to adults in face- and non-face images was that there may be category-specific developments that affect the ERP response to varying MI levels. To be more precise, we hypothesized while adults would show gradient-like sensitivity to face fragment informativeness (MI level) at the amplitudes of the P100 and N170 components, children may not yet have developed the same sensitivity to fragment informativeness, and thus show a reduced effect of MI level. Besides this overarching developmental hypothesis, we also expected to largely replicate the main results of Harel et al. [32], in which ERPs were also recorded in response to face and non-face fragments that varied in MI.

Our results from adult participants largely replicated the effects reported by Harel et al. [32]. In both studies, varying levels of MI had little to no impact on the amplitude of the P100. The lack of measurable effects of MI on the P100 is an important indicator that low-level differences between the various groups of fragments are unlikely to be the underlying cause of effects observed later in the waveform. The effects observed at later components (the N170 in the current study and a component that Harel et al. refer to as the N270) are, therefore, more likely to reflect higher-level, face-selective processes. In both cases, increasing levels of MI led to higher amplitudes (more negative) ERP responses at downstream components, which is consistent with multiple prior studies demonstrating that less face-like images tend to elicit lower amplitude N170 responses [39,45]. Additionally, while our focus in this study was on the P100 and N170 components, we do in fact observe a negative deflection that appears to correspond well to the N270 component that Harel et. al. reported. Moreover, this component also appears to exhibit a gradient-like response to varying MI levels that is consistent with their results: Increasing fragment informativeness leads to larger amplitudes. Unlike their results, however, we did not observe effects of fragment informativeness on non-face fragments (cars) either at our target components or the N270 component. Our data from adult participants thus confirms that the visual system is sensitive to MI in face fragments and not solely due to uncontrolled low-level variation in properties like brightness, contrast, or size. This is further supported by the behavioral data obtained from adults, which also suggests that varying the MI level in partial face fragments affects the extent to which such partial images are categorized as faces.

Considering our data from children in this context leads to several interesting observations. One particularly conspicuous feature of our results is the apparent absence of an N270 component in both younger and older children. While our adults exhibit a clear positive and negative deflection following the N170, neither group of children has any such pattern. Harel et al. proposed that their N270 component may reflect some specific response to the presentation of image fragments, and if this is the case, our results suggest that children’s response to such fragments must change dramatically between middle childhood and adulthood. Even if the N270 is not really a fragment-specific response, ERP components following the N170 have been linked to categorization processes (e.g., race and age [54,55,56] and the processing of face familiarity [57] or identity [58]). Another interpretation of our results then is that these categorization processes may be substantially disrupted when relatively impoverished stimuli like these are used. In either case, the clear sensitivity to MI in adults’ downstream response to face fragments is in stark contrast to the absence of such a response in children. Further exploration of why these downstream components are missing in our child participants is warranted and may lead to interesting results regarding how different levels of categorization are affected by varying the visual information available in face stimuli.

Another intriguing difference between child and adult ERPs in our study is the nature of the interaction between stimulus category, MI level, and age group. Briefly, while adults’ sensitivity to face MI levels is reflected in their N170 amplitudes, young children’s sensitivity to face MI appears to be reflected in their P100 amplitudes. Older children appear to be in a sort of transition phase, exhibiting no obvious sensitivity to MI at either component. This pattern of results indicates that there may be a substantial shift in how faces are processed during childhood such that information about ‘face-ness’ is manifest at different stages of visual processing. A similar pattern of results was reported by Peters et al. [42] who observed that the face inversion effect (FIE) was evident at different components as a function of development in a task where they varied the spatial frequency content of face images presented to adult, adolescent, and child participants. While young children exhibited an FIE at the P100 (similar to the sensitivity to MI we observed at the P100), adults exhibited an FIE at the N170 (similar to our results), while adolescents exhibited effects at both components. Their conclusion was that the effects observed at the P100 in young children may have reflected a more superficial, less face-specific response to upright and inverted faces. Another possibility, articulated by Taylor et al. [48], is that the P100 reflects basic categorization of a face stimulus as an upright face, while the N170 reflects more complex processes related to holistic or configural processing. In this context, our results could be interpreted to mean that young children do not automatically process low-MI face fragments as faces (leading to an effect of MI at the P100) and also may not process any face fragment holistically or configurally. By contrast, adults may easily categorize all face fragments as being face-like, but vary in their application of face-specific processing to low vs. high-MI fragments. In either case, we suggest that middle childhood may be characterized by an important developmental shift in which neural processes are tuned to the diagnosticity of both low-level and mid-level face features. Our behavioral results from both groups of children provide an important context for this interpretation, however. In particular, we note that both groups of children had lower accuracies across all face MI levels than adults, which suggests that, in general, these partial fragments of images are less likely to be perceived as faces. This could reflect the changes in neural tuning that we have discussed above, but could also indicate that there are broader visual processes related to face categorization that are also not yet mature or not sensitive to the features that tend to be contained in face fragments. At present, we would suggest that the main contribution of the current study is that we see complementary behavioral and neural evidence that partial face fragments with varying levels of MI elicit different responses from observers across middle childhood. Whether or not we should consider the change in ERP responses as the primary developmental change in face categorization or not is not clear from our data. Overall, we would instead emphasize that children’s continuing refinement of face representations to support face categorization is reflected in both their behavioral responses and different portions of their ERP responses throughout the middle childhood years.

More broadly, our results are also consistent with a number of previous results demonstrating that in terms of information use, the development of face recognition is relatively slow. While some aspects of face recognition may be mature fairly early in childhood (e.g., the composite face effect, e.g., [29]), in multiple studies measuring how children use diagnostic vs. non-diagnostic information for various tasks, children’s information use is not adult-like until after age 10 or beyond. For example, both behavioral [11,12,13] and neural results [40] describing sensitivity to horizontal orientation energy in face images suggest that while children may show some early tendencies towards an adult-like horizontal bias, there is an ongoing refinement of this bias throughout middle childhood. Likewise, children’s use of low spatial frequencies vs. high spatial frequencies in multiple behavioral tasks [6] and ERP studies [42] also changes gradually during this developmental period. We suggest that these results, along with our own, point towards a developing visual system that continues to modify its vocabulary of visual features for face recognition between the ages of 5–10 years and possibly beyond. In contrast with theories of the development of face recognition that emphasize ways in which children’s face recognition is adult-like in some ways [59], our results are consistent with the view that there is an ongoing refinement of the mechanisms used to recognize and detect faces. Critically, this refinement occurs at multiple levels of complexity, affecting how visual features at many stages of processing are used to detect and recognize face images.

Our results suggest several avenues for further investigation. In particular, it would be interesting to examine the P100-N170 shift in sensitivity to MI levels using tools that support better localization of the underlying neural signals. Multiple neural loci (including the occipital face area [60], the fusiform face area [61], and the superior temporal sulcus [62]) exhibit varying sensitivities to different aspects of face appearance [63]. For example, the occipital face area appears to be sensitive to the presence of face parts, but not their spatial configuration [64], while later stages of face processing are sensitive to the arrangement of face parts into a typical pattern. The developmental shift that we have observed here with regard to MI sensitivity may thus also be related to developmental trajectories specific to those regions, which is difficult to observe given the poor spatial resolution of EEG and ERP. The use of either fNIRS or fMRI to examine MI sensitivity within specific face-sensitive areas would thus be an important complement to the present work. Besides characterizing how MI sensitivity may be linked to specific neural loci, it would also be useful to characterize how MI sensitivity is linked to observers’ sensitivity to other diagnostic visual features for face recognition. At the level of individual participants, for example, do changes in sensitivity to spatial frequency or orientation precede changes in sensitivity to mid-level or high-level features? Alternatively, is there a gradient of development in the opposite direction, or no such consistent relationship at all? Either way, examining possible dependencies between features of different levels of complexity would be an important way to establish how children ultimately acquire an adult-like representation of facial appearance that allows them to efficiently and accurately detect and recognize faces.

## 5. Conclusions

Our results revealed that the developing visual system’s sensitivity to the mutual information available in fragments of face images changes during childhood. Gradient-like sensitivity to face fragment informativeness shifts from early to later components between early childhood and adulthood, potentially reflecting an important cortical reorganization of face-processing mechanisms. Considering how faces are recognized using mid-level features is an important complement to prior work examining sensitivity to visual features that are either more or less complex. Our results contribute to an emerging characterization of the development of face recognition that includes the gradual refinement of the vocabulary of recognition, which is reflected in changing neural responses to diagnostic visual information.

## Figures and Tables

**Figure 1 brainsci-09-00154-f001:**
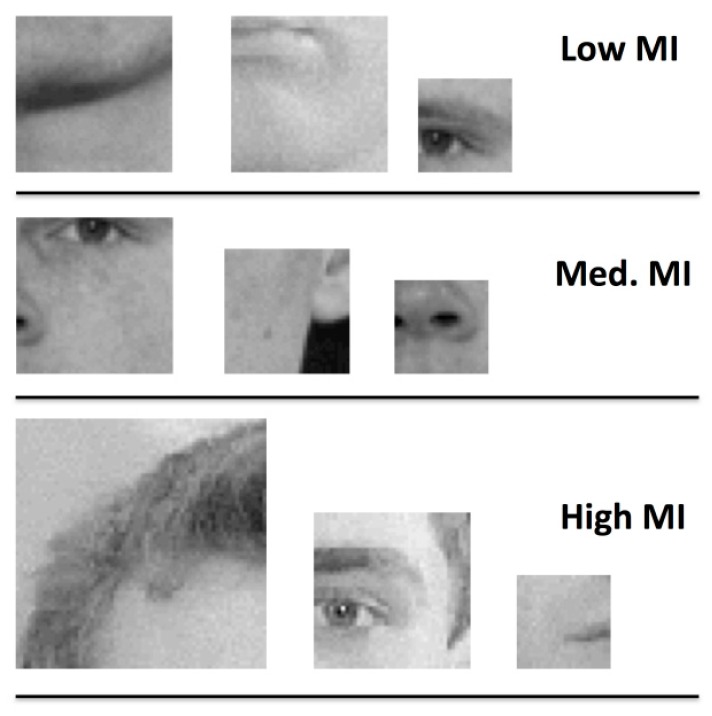
Examples of face fragments with low, medium, and high mutual information. Fragments in both categories span a range of sizes and depict portions of different discrete face parts.

**Figure 2 brainsci-09-00154-f002:**
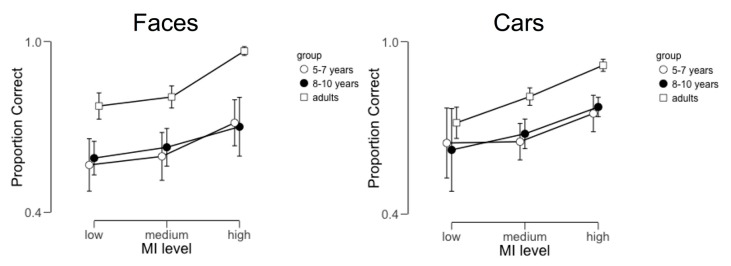
The average proportion correct across participants as a function of age, target category (faces or cars) and mutual information (MI) level (low, medium, and high). Error bars represent 95% credible intervals.

**Figure 3 brainsci-09-00154-f003:**
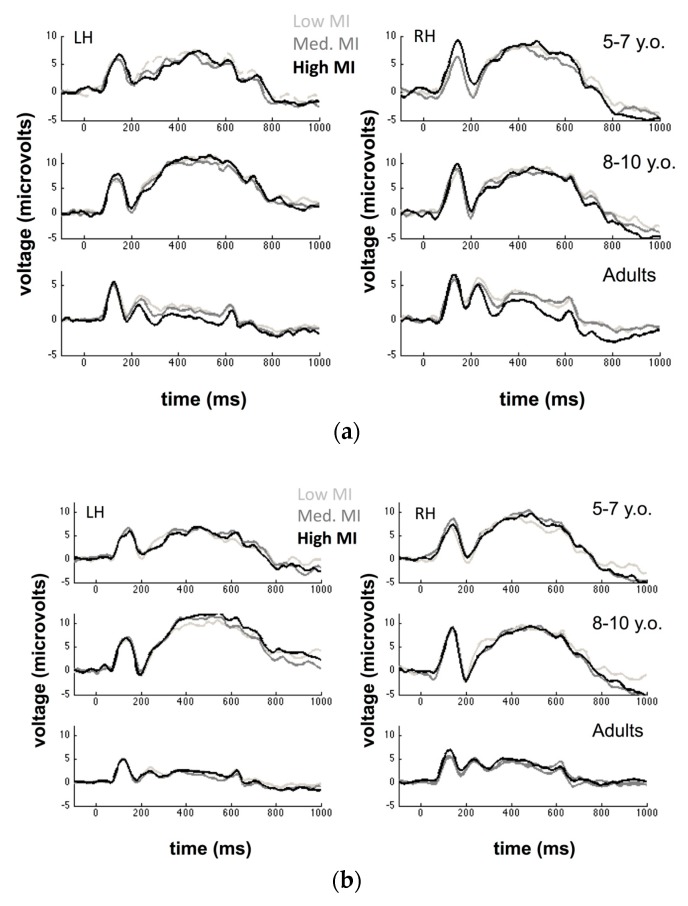
(**a**) The grand average event-related potentials (ERPs) (Face fragments) across participants measured over the left hemisphere and right hemisphere electrodes. Note that both panels of the adult data are scaled differently along the *y*-axis to make it easier to see the different condition waveforms. (**b**) The grand average ERPs (Car fragments) across participants measured over left hemisphere and right hemisphere electrodes. y.o.: year-old.

**Figure 4 brainsci-09-00154-f004:**
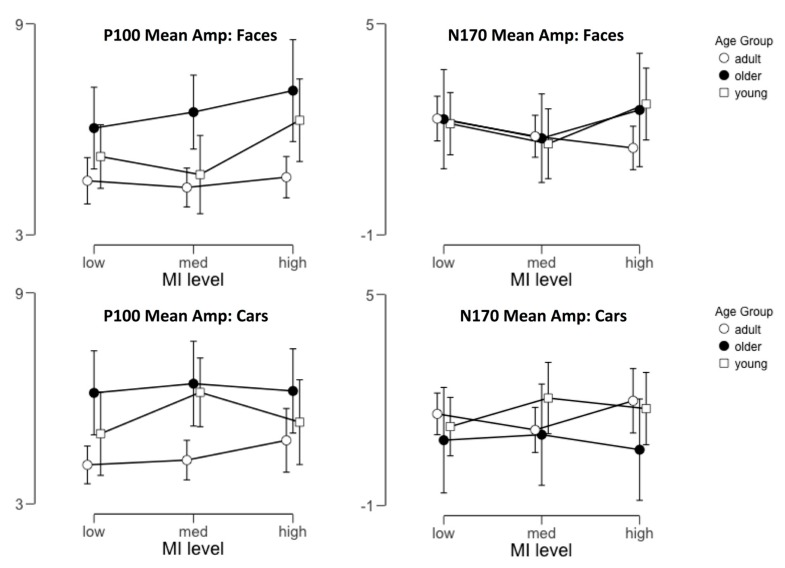
The average mean amplitude across participants for the P100 (**left**) and N170 components (**right**) in response to both faces (**top row**) and non-faces (**bottom row**). In each case, we have extracted the mean amplitude for each participant’s target component across condition and plotted the aggregate data as a function of the target category and MI level. Error bars indicate 95% confidence intervals.

**Table 1 brainsci-09-00154-t001:** The full ANOVA table for the analysis of within-subject effects on the mean P100 amplitude.

Within-Subjects Effects										
	Sum of Squares		*df*		Mean Square		*F*		*p*	
Category	1.529		1		1.529		0.516		0.476	
Category × Age Group	3.440		2		1.720		0.580		0.564	
Residual	136.331		46		2.964					
MI level	29.562	ᵃ	2	ᵃ	14.781	ᵃ	4.244	ᵃ	0.017	ᵃ
MI level × Age Group	3.506	ᵃ	4	ᵃ	0.877	ᵃ	0.252	ᵃ	0.908	ᵃ
Residual	320.421		92		3.483					
Hemisphere	116.881		1		116.881		2.335		0.133	
Hemisphere × Age Group	6.672		2		3.336		0.067		0.936	
Residual	2302.144		46		50.047					
Category × MI level	23.957		2		11.978		3.655		0.030	
Category × MI level × Age Group	37.705		4		9.426		2.876		0.027	
Residual	301.536		92		3.278					
Category × Hemisphere	3.112		1		3.112		2.097		0.154	
Category × Hemisphere × Age Group	5.540		2		2.770		1.866		0.166	
Residual	68.268		46		1.484					
MI level × Hemisphere	20.783		2		10.392		5.672		0.005	
MI level × Hemisphere × Age Group	14.502		4		3.626		1.979		0.104	
Residual	168.548		92		1.832					
Category × MI level × Hemisphere	1.899		2		0.949		0.322		0.726	
Category × MI level × Hemisphere × Age Group	23.223		4		5.806		1.968		0.106	
Residual	271.355		92		2.950					

Type III Sum of Squares; ᵃ Mauchly’s test of sphericity indicates that the assumption of sphericity is violated (*p* < 0.05).

**Table 2 brainsci-09-00154-t002:** The full ANOVA table for the analysis of within-subject effects on the mean N170 amplitude.

Within-Subjects Effects										
	Sum of Squares		*df*		Mean Square		*F*		*p*	
Category	73.860		1		73.860		13.464		<0.001	
Category × Age Group	33.300		2		16.650		3.035		0.058	
Residual	252.352		46		5.486					
MI level	8.095	ᵃ	2	ᵃ	4.048	ᵃ	0.803	ᵃ	0.451	ᵃ
MI level × Age Group	11.274	ᵃ	4	ᵃ	2.819	ᵃ	0.559	ᵃ	0.693	ᵃ
Residual	463.544		92		5.039					
Hemisphere	5.603		1		5.603		0.111		0.740	
Hemisphere × Age Group	185.081		2		92.541		1.835		0.171	
Residual	2319.340		46		50.420					
Category × MI level	12.970		2		6.485		1.663		0.195	
Category × MI level × Age Group	36.809		4		9.202		2.359		0.059	
Residual	358.851		92		3.901					
Category × Hemisphere	0.009		1		0.009		0.005		0.944	
Category × Hemisphere × Age Group	13.080		2		6.540		3.539		0.037	
Residual	85.019		46		1.848					
MI level × Hemisphere	8.952		2		4.476		1.382		0.256	
MI level × Hemisphere × Age Group	14.313		4		3.578		1.105		0.359	
Residual	297.972		92		3.239					
Category × MI level × Hemisphere	3.836	ᵃ	2	ᵃ	1.918	ᵃ	0.870	ᵃ	0.422	ᵃ
Category × MI level × Hemisphere × Age Group	8.691	ᵃ	4	ᵃ	2.173	ᵃ	0.986	ᵃ	0.419	ᵃ
Residual	202.762		92		2.204					

Type III Sum of Squares; ᵃ Mauchly’s test of sphericity indicates that the assumption of sphericity is violated (*p* < 0.05).

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
