# Peer review of "Neural Sensitivity to Mutual Information in Intermediate-Complexity Face Features Changes during Childhood"

_brainsci, 2019, doi:10.3390/brainsci9070154_

Reviewer 1 Report

The authors present interesting information about children and adult neural sensitivity / response for face recognition and decisions. The authors present the term mutual information which is use to develop children neural sensitivity mid-level face features and categorization of face and non-face images. In this regards, authors use P100 and N170 components. 

The manuscript is well written, has important clinical message, and should be of great interest to the readers. However, the experimental and result part need to be revised. The author didn’t clearly describe the term MI and some of figures. There should be some mathematically equations & information in this manuscript instead of huge theory. Overall, it is an important study. This paper has a potential to be accepted, but some important points have to be clarified or fixed.

1.    It is unclear about the definition the term “Mutual Information”. In the field of computer science, we have a strict mathematical definition about “Mutual Information”. But in this paper, the author does not formally define this term. Is MI in this study defined as the same way as we did in computer science? Or is there another definition of MI which is more popular in the field of cognitive science and psychology? What is formal way (in terms of math equation) to calculate MI value? Please also give some toy example to illustrate how to measure or compute MI given a set of experimental data.

2.     Stimuli part is not clear. Provide detail about images of face data. How many number in total are there for face/cars images. How did you divide fragments of faces/cars into the three MI level?  Based on what theory or previous study can you be assured that such classification is correct?

3.    Figure 3. Is not clear to understand. Explain it precisely.

Author Response

The authors present interesting information about children and adult neural sensitivity / response for face recognition and decisions. The authors present the term mutual information which is use to develop children neural sensitivity mid-level face features and categorization of face and non-face images. In this regards, authors use P100 and N170 components. 

The manuscript is well written, has important clinical message, and should be of great interest to the readers. However, the experimental and result part need to be revised. The author didn’t clearly describe the term MI and some of figures. There should be some mathematically equations & information in this manuscript instead of huge theory. Overall, it is an important study. This paper has a potential to be accepted, but some important points have to be clarified or fixed.

Thank you for your comments. Below, we address the various points you have raised in turn, identifying how we have changed the text in response.

1.     It is unclear about the definition the term “Mutual Information”. In the field of computer science, we have a strict mathematical definition about “Mutual Information”. But in this paper, the author does not formally define this term. Is MI in this study defined as the same way as we did in computer science? Or is there another definition of MI which is more popular in the field of cognitive science and psychology? What is formal way (in terms of math equation) to calculate MI value? Please also give some toy example to illustrate how to measure or compute MI given a set of experimental data.

Yes, we are using a definition of mutual information which is in agreement with the use of the term in other disciplines. Specifically, mutual information (or MI, as we write in the manuscript) is defined for our face and car fragments based on the information that a fragment provides regarding the target category. The formal definition for MI in the context of image fragments is given in the previous studies we have cited, but we also briefly reproduce the relevant equations in the revised text along with a short description of how MI is estimated for a set of fragments and images.

2.     Stimuli part is not clear. Provide detail about images of face data. How many number in total are there for face/cars images. How did you divide fragments of faces/cars into the three MI level?  Based on what theory or previous study can you be assured that such classification is correct?

We apologize for the confusion. We have edited the text to more carefully describe the number of stimuli in each condition. The different MI levels were selected by partitioning the observed MI values into non-overlapping bins. The correctness of this measure has to be considered two ways: (1) We computed MI values according to the equations and procedures described in response to the reviewer’s first point. On this basis, the MI values are meaningful because they reflect the information we gain about a target category for observing a specific fragment. (2) Just because this value has a computational meaning doesn’t imply that it also has a perceptual meaning. This latter question is the focus of the current study.

3.    Figure 3. Is not clear to understand. Explain it precisely. 

We apologize for the lack of clarity – we have attempted to explain the figure more clearly in the revised caption. Note that in the revised manuscript this is now Figure 4.

Reviewer 2 Report

Balas et al. investigate the erp to partial images of faces and cars that vary in MI. Overall this is an interesting manuscript, providing an unique contribution to the face literature, but there are major and minor issues to address, presented below. I did not dig into the discussion, hoping we can first resolve what I see as issues in the analysis.

1) the paragraph starting at line 110 - needs further development/clarification of a) what the P100 and N170 (and perhaps the n270?) are thought to represent, and b) how the partial stimuli are expected to drive these ERP components in the context of faces vs. cars.

2) the manuscript lacks behavioral data - it is important to know the timeline for categorization in each condition (RT) and the quality of the responses (their accuracy). please add these.

3) at line 247, it would be nice if the time windows were the same or a justification were provided as to why those particular time ranges are used. I am not an expert in the child face literature, but perhaps there is a citation that would justify moving the window or provide some overtly more objective criteria.

4) in figure 2, a major concern is the apparent lack of a p100 (having an explanation for this absence would be helpful, see pt 1 above). As a comparison, Miki et al. (2015) (doi: https://doi.org/10.3389/fnhum.2015.00263) showed faces, inverted faces, and partial faces (eyes) and generated a strong p100 in each case. Not to dictate or ridicule your experimental design, but a positive control of faces generating a p100 (and something of a negative control in terms of images of entire cars) for comparison in the same session as the other stimuli would have been exceedingly helpful here. As it is, the analysis of the p100 magnitudes seems specious in the absence of a visible p100 - given an apparent lack of a p100, what can a significant main effect or interaction mean regarding the p100?

5) also in figure 2, if you are plotting negative components upward it would be helpful to have negative units on the axis going up instead of down (if negative erps go up, negative should be up on the axis label). While you're in there, it would be nice if the axes had the same magnitude, even if the adults ERPs are smaller than the kids.

6) at line 129, it would be helpful to explain that the negativity at 270ms is thought to be a component linked to Harel as explained in line 330.

7) as well, it may be worth introducing the n270 in the abstract, expanding its discussion the intro, and analyzing a 270ms window’s average amplitude as you seem to speculate about it in the discussion and it is more readily apparent in the erps.

Author Response

Balas et al. investigate the erp to partial images of faces and cars that vary in MI. Overall this is an interesting manuscript, providing an unique contribution to the face literature, but there are major and minor issues to address, presented below. I did not dig into the discussion, hoping we can first resolve what I see as issues in the analysis.

Thank you very much for your comments. Below we address each of the points you raised in turn, describing how we have edited the text to address your comments in each case.

1)     the paragraph starting at line 110 - needs further development/clarification of a) what the P100 and N170 (and perhaps the n270?) are thought to represent, and b) how the partial stimuli are expected to drive these ERP components in the context of faces vs. cars.

We have edited the text to make this more clear. The P100 is largely accepted as an index of low-level visual processing – this component is sensitive to the luminance, contrast, and spatial frequency content of stimuli. However, it also exhibits  sensitivity to upright faces as opposed to inverted faces, however, which is a hallmark of face-specific processing. The N170 is widely established as a face-sensitive component: The N170 is generally much larger in amplitude in response to faces compared to other object classes and is sensitive to face inversion, face contrast negation, and other image manipulations that affect face recognition performance. Both of these components thus have known response properties that suggest they contribute to face-specific visual processing. Finally, the N270 was identified by Harel et al., in a prior study that used face fragments. This component was found to exhibit a gradient-like response to varying MI, which makes it an important consideration for our study as well. While it's functional role is less well-understood than either the P100 or N270, it may reflect an independent response to partial face images.  In general, we expect that all of these components should exhibit more robust responses the more face-like an image fragment is. With regard to cars, to the extent that these are truly face-sensitive components, we would expect that they may be less sensitive to varying car-ness across image fragments.

2)     the manuscript lacks behavioral data - it is important to know the timeline for categorization in each condition (RT) and the quality of the responses (their accuracy). please add these.

We have included an analysis of our participants’ accuracy data in the revised text. The response latency data is complicated by the fact that participants were asked to withhold behavioral responses until images disappeared from view, so these latencies likely do not reflect the true speed of their response to these images. Further, young children frequently took breaks throughout the experimental session by pausing between trials, which makes it even more difficult to rely on these data to describe the speed of processing. As such, we have omitted an analysis of the response latency data.

3)     at line 247, it would be nice if the time windows were the same or a justification were provided as to why those particular time ranges are used. I am not an expert in the child face literature, but perhaps there is a citation that would justify moving the window or provide some overtly more objective criteria.

We understand the reviewer’s concern. The key issue here is that the latency and width of the target components is not the same across our age groups. Kuefner et al., who we cite in the revised text, reported similar shifts of component latency and morphology across the same age range as we use here, which we think is a useful precedent for our approach. Faced with components that vary across age in terms of their latency and width, we see no easy solution save for adjusting time windows to track this variability across groups. We describe this issue in more detail in the revised text.

4)     in figure 2, a major concern is the apparent lack of a p100 (having an explanation for this absence would be helpful, see pt 1 above). As a comparison, Miki et al. (2015) (doi: https://doi.org/10.3389/fnhum.2015.00263) showed faces, inverted faces, and partial faces (eyes) and generated a strong p100 in each case. Not to dictate or ridicule your experimental design, but a positive control of faces generating a p100 (and something of a negative control in terms of images of entire cars) for comparison in the same session as the other stimuli would have been exceedingly helpful here. As it is, the analysis of the p100 magnitudes seems specious in the absence of a visible p100 - given an apparent lack of a p100, what can a significant main effect or interaction mean regarding the p100?

We apologize if the original text or figures were confusing. We did in fact observe a prominent P100 component in all three age groups and defined our time windows based on this component’s latency. Respectfully, in each subplot of Figure 3, the P100 we are analyzing is the first positive deflection in each waveform, which has a latency that varies somewhat across age groups and conditions but is generally a bit later than 100ms. In general, we would therefore say that our P100 component is shifted somewhat later in the timecourse compared to prior work, but this is typical of child ERP data and w.r.t adults, may be a consequence of the fragmentary images we used.

5)     also in figure 2, if you are plotting negative components upward it would be helpful to have negative units on the axis going up instead of down (if negative erps go up, negative should be up on the axis label). While you're in there, it would be nice if the axes had the same magnitude, even if the adults ERPs are smaller than the kids.

To clarify, we are not plotting negative components upward – we are plotting positive components upward. This may be the source of the confusion regarding the P100 in our data. Regarding the scale of the axes, we have opted to highlight in the caption that the axes are scaled different across age groups when this has been applied. Resizing these to a common scale makes it extremely difficult to see the effects in the adult data (we tried, and the resulting figure was very hard to interpret), so we think it is perhaps easier for the reader to retain different scaling so long as this is clearly communicated in the text.

6)     at line 129, it would be helpful to explain that the negativity at 270ms is thought to be a component linked to Harel as explained in line 330.

We agree and have included this in the edited text.

7)     as well, it may be worth introducing the n270 in the abstract, expanding its discussion the intro, and analyzing a 270ms window’s average amplitude as you seem to speculate about it in the discussion and it is more readily apparent in the erps.

We agree that the earlier discussion of this component is useful. With regard to analyzing it across age groups, however, the problem we face is the profound lack of an observable N270 in either of our child age groups. As the reviewer observed in an earlier comment, it is hard to meaningfully measure a component when it is not clearly evident in the ERP waveform. In the case of the N270, we simply do not see an obvious N270 in 2 of our 3 participant groups, and so must decline to provide such an analysis in the current study.

Thank you again for your comments regarding the initial submission. We agree that these revisions strengthen the manuscript considerably and we hope we have adequately addressed the points you raised.

Round  2

Reviewer 2 Report

Thank you for the clarifications and revisions – they’ve been very helpful for reading the manuscript. In the revisions, a few typos:

line 261 – “of” should be “or”

line 286 – “We continue by describing our” should be deleted

line 288 – “out” should be “our”

The issues raised in the first round of review have been addressed and I have no problems through the first two paragraphs of the discussion with regard to the interpretation of MI and P100, N170 and N270.

The behavioral data bring forth a concern with respect to the results/discussion, which could be satisfied with one further analysis or a strong caution in the text. Given the behavioral data presented in Figure 2, it seems another explanation is plausible regarding the sensitivity to fragment informativeness and effects of MI level on ERPs in childhood that is discussed particularly in the paragraph from 418-433, but also from 434-474. Specifically, the issue is that there is significantly lower categorization accuracy in the child groups that borders on guessing (obviously this will vary from child to child, and stimulus to stimulus). If the children are guessing sometimes (or some children were guessing consistently), and the N170 and N270 reflect higher-level, face-selective processes as suggested at line 406-407, then the results are as would be expected – if the children’s visual systems do not identify something as a face, the face-selective mechanisms would not engage. Without full or partial face stimuli for comparison that do elicit an N170 and N270, or evidence that the children can definitely identify such stimuli as faces, it is hard to know if this is an ERP that develops/shifts in the intervening years or if it is just absent in these data because the children were not processing the MI-containing images as faces.

A reanalysis of correct vs. incorrect trials that shows there is still no evidence of such components would alleviate my concerns, but if this is prohibitive, then the manuscript would at least benefit from a discussion of the possibility that the components could be absent/different in children due to higher rates of guessing. I recognize that this possibility still arises with the suggested re-analysis (sometimes they will have guessed correctly), but at least it removes some of the noise generated by incorrect trials – particularly with the adults performing significantly better in all conditions.

Author Response

We would like to thank the reviewer for their additional comments regarding our manuscript. We were pleased to see that they were generally positive regarding the revisions we made following the first round of review and describe below how we have edited the text in response to their most recent comments.

We agree with the reviewer that the behavioral data does raise an important issue regarding how we should interpret the results we describe regarding our ERP data. We also agree that it could certainly be the case that children’s inability to process faces with low MI as face stimuli could underlie the difference between child and adult responses at the neural level. We think that the important contribution of our study relies in large part on the fact that in general, children’s ERP data is not completely determined by their behavioral responses. Another way to put this is to say that it was certainly possible that this could have come out differently: Even chance-level behavioral responses to these stimuli could have elicited strong effects at the N170, at the very least because the EEG signatures of face processing could be more sensitive than behavioral measurements. Here, this is not what we observed, so we agree with the the reviewer that it is important to point out that we cannot distinguish between an effect that is the result of the N170/N270 components themselves changing with development, or a more general developmental change that affects how partial face stimuli are categorized. In the revised text, we have edited the sections that the reviewer pointed out to emphasize this point, so our readers can interpret our ERP results appropriately in the context of our behavioral data. While we agree that it would be ideal to include a breakdown of correct/incorrect trials for child participants in this study, the number of trials in each of those conditions would be sufficiently low that we don’t think they would be a useful basis for resolving this important issue. As such, we have adopted the reviewer’s suggestion to edit the text of our discussion section to address this point.

Besides these revisions, we have also corrected the more minor errors that the reviewer pointed out. We would like to thank the reviewer again for their commentary, and hope that our responses adequately address the remaining issues they raised in this round of review.